# Direct liquid transmission of sound has little impact on fermentation performance in *Saccharomyces cerevisiae*

Rachel Benitez[1☉], Alastair Harris[1☉], Evie Mansfield[1], Pat Silcock[2], Graham Eyres[2], Silas G. Villas-Bôas[3], Andrew Jeffs[1], Austen R. D. Ganley[1]*

1 School of Biological Sciences, University of Auckland, Auckland CBD, New Zealand, New Zealand,
2 Department of Food Science, University of Otago, Dunedin, New Zealand, 3 Luxembourg Institute of Science and Technology, Z.A.E. Robert Steichen, Luxembourg, Luxembourg

☉ These authors contributed equally to this work.
* a.ganley@auckland.ac.nz

**Data Availability Statement:** All data generated or analyzed during this study are included in this paper and its Supporting Information files, except the sound .wav files, which are available through

## Abstract

Sound is a physical stimulus that has the potential to affect various growth parameters of microorganisms. However, the effects of audible sound on microbes reported in the literature are inconsistent. Most published studies involve transmitting sound from external speakers through air toward liquid cultures of the microorganisms. However, the density differential between air and liquid culture could greatly alter the sound characteristics to which the microorganisms are exposed. In this study we apply white noise sound in a highly controlled experimental system that we previously established for transmitting sound underwater directly into liquid cultures to examine the effects of two key sound parameters, frequency and intensity, on the fermentation performance of a commercial *Saccharomyces cerevisiae* ale yeast growing in a maltose minimal medium. We performed these experiments in an anechoic chamber to minimise extraneous sound, and find little consistent effect of either sound frequency or intensity on the growth rate, maltose consumption, or ethanol production of this yeast strain. These results, while in contrast to those reported in most published studies, are consistent with our previous study showing that direct underwater exposure to white noise sound has little impact on *S. cerevisiae* volatile production and sugar utilization in beer medium. Thus, our results suggest the possibility that reported microorganism responses to sound may be an artefact associated with applying sound to cultures externally via transmission through air.

## Introduction

The effects of environmental stimuli, such as temperature, oxygen and nutrient availability, on microbial growth and behaviour are well known and are carefully managed in commercial applications [1–3]. In contrast, sound as an environmental stimulus has received less research attention and receives scant attention in commercial applications. Published results indicate that audible sound (20 Hz– 20 kHz) [4] stimulation can directly affect growth and other

Figshare (https://auckland.figshare.com/articles/media/Sound_wav_files_use/20103734; DOI: 10.17608/k6.auckland.20103734).

**Funding:** This work was supported by a Smart Ideas grant from the New Zealand Ministry for Business, Innovation, and Employment [UOAX1713; https://www.mbie.govt.nz/science-and-technology/science-and-innovation/funding-information-and-opportunities/investment-funds/endeavour-fund/] to ARDG, AJ, SGV-B, PS and GE. The funders had no role in study design, data collection and analysis, decision to publish, or preparation of the manuscript.

**Competing interests:** The authors have declared that no competing interests exist.

processes in microbial cultures. For example, audible sound has been reported to affect gram-negative bacteria, by increasing cell viability, colony formation and biomass production [5–9], and increase growth, antibiotic susceptibility, and endospore germination [10]. Audible sound stimulation has also been reported to increase growth, antibiotic susceptibility and the production of industrially relevant compounds in the yeast *Candida albicans*, and to increase both growth rate and total biomass in microalgae [8, 11, 12]. However, we still do not know the mechanistic basis for how the effects of sound on microbes are mediated.

Sound is a mechanical wave that exerts physical forces on culture media and is characterized by the frequency (tone or pitch), which is measured in hertz (Hz), and the amplitude (loudness or intensity), which is measured in decibels (dB), a relative unit of measure for sound pressure [13]. The pressure of sound is measured in relation to a reference pressure level determined in micro-Pascals (μPa). For sound transmission in air this reference is standardized to 20 μPa, the absolute threshold of perception in humans for a sound frequency of 1,000 Hz, whereas the reference for underwater sound is set at 1 μPa by convention [14]. The medium through which sound is transmitted influences the resulting sound waves. For example, sound transmitted through liquids travels four times faster than through air, and is generally subject to less attenuation (loss of energy) so can be transmitted for longer distances [15]. Most studies of the response of microbial organisms to audible sound have used sound sources, such as loudspeakers, to transmit sound through air toward microbial liquid cultures held within hard-walled vessels, such as glass flasks. Marked density differentials between air, liquid, and vessel wall material likely result in sound transmission losses [16], with the level of transmission loss likely influenced by the frequency [17]. Consequently, the sound entering the cultures and affecting the microbes may be substantially different to that being transmitted. Furthermore, the prevalence of sound in many laboratories makes high-quality sound proofing necessary to reduce the influence of background sound. Conducting sound experiments in an anechoic chamber, which is a specifically designed acoustic room that limits background noise by absorbing or reflecting sound, can be an effective way to limit background sound [18]. *Saccharomyces cerevisiae* is widely used in the production of alcoholic beverages [19] through fermentation. It is the main organism for production of ale (top-fermenting) beers, and it demonstrates clear evolutionary domestication to beer fermentation [20, 21]. Despite the clear industrial relevance of *S. cerevisiae* fermentation, relatively few studies have investigated how sound affects *S. cerevisiae* fermentation properties. Three studies report that sound increases *S. cerevisiae* growth rate, although they each differ in the sound treatment applied [11, 22, 23]. There is also an indication that sound can reduce the duration of the *S. cerevisiae* exponential growth phase during fermentation, increase alcohol production, and alter the metabolic flux through many pathways [22]. However, these previous studies all transmitted sound through air at liquid cultures, and the sound conditions to which the yeast were exposed within the culture vessels were not measured. To overcome the limitation of transmitting sound through air into liquid, we recently developed an experimental system that applies sound under water into liquid cultures [24]. Surprisingly given previous studies, we found little impact of audible sound application on sugar utilization and volatile organic compound production when *S. cerevisiae* was fermenting a beer medium [24]. Here we used a common *S. cerevisiae* ale strain fermenting a simple minimal medium to test the effects of different sound frequencies and intensities within the audible range using our underwater experimental sound system in an anechoic chamber on *S. cerevisiae* fermentation parameters commonly reported to respond to sound. Overall, consistent with our previous study, we found little evidence for an effect of audible white noise sound on the fermentation performance of *S. cerevisiae*.

## Results

The purpose of this work was to determine the effect of frequency and intensity of audible sound on commonly-reported *S. cerevisiae* fermentation parameters. We decided to perform this work using a simple minimal medium to reduce potential confounding effects of the complex beer medium we used in our previous paper [24], and in an anechoic chamber to maximally reduce extraneous sounds. Our experimental system applies sound through an underwater speaker submerged in a water barrel, with the culture vessel also submerged in the same barrel (Fig 1), to overcome sound transmission issues caused by density differences between air, the hard culture vessel walls, and the liquid medium. This, in principle, means we can accurately transmit known sound profiles to *S. cerevisiae* fermentations.

### Validation of sound transmission into culture vessels

We first wanted to confirm that the sound properties being transmitted by the underwater speaker through the water in the barrel are those experienced by the culture medium. As culture vessel walls have a different density to that of the water, we tested three different types of fermentation vessel (glass round bottom flask, hard-plastic Nalgene bottle, and soft plastic IV

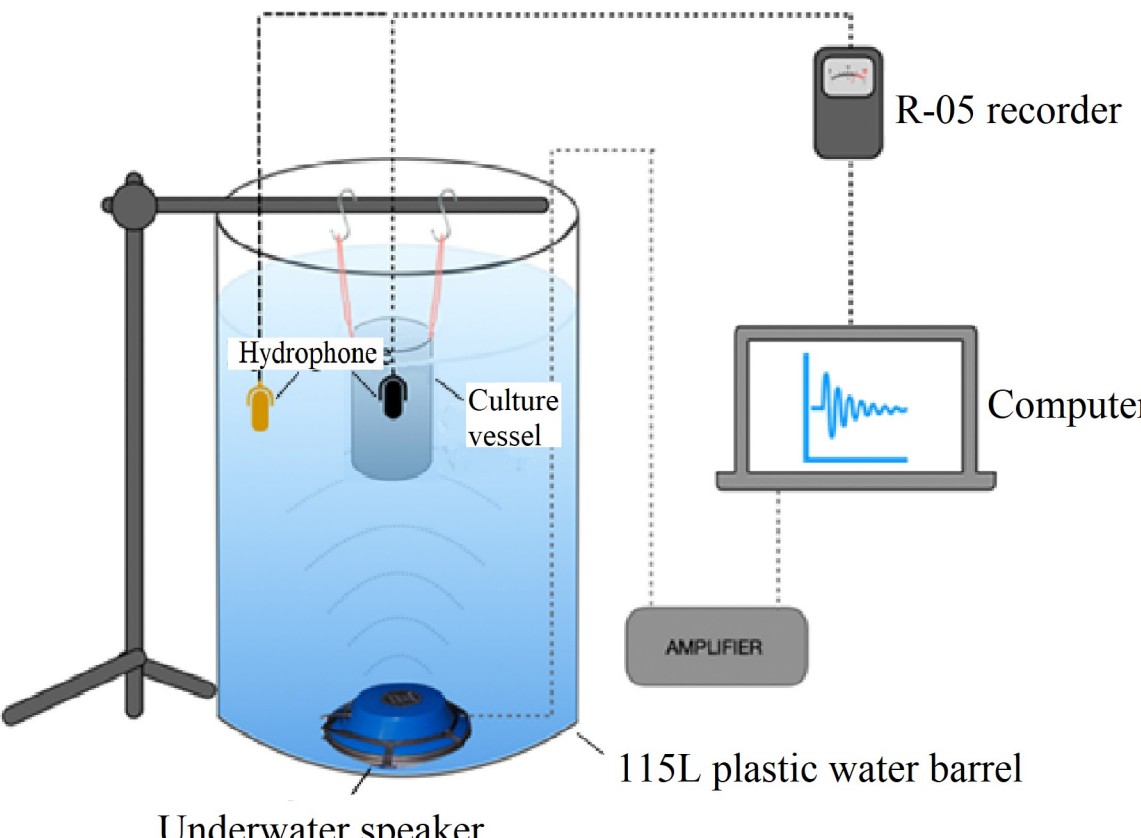

**Fig 1. Set-up for sound transmission testing.** Culture vessels were suspended by rubber bands from a microphone stand to acoustically isolate the vessel and submerged underwater in a 115 L plastic barrel filled with freshwater. Sound source files were played from a computer, amplified, then used to drive an underwater speaker at the bottom of the barrel. Transmitted sound within the barrel was measured with hydrophones and R-05 sound recorder, then processed in MATLAB (S1 Appendix). During sound transmission testing, a hydrophone was suspended within the culture vessel, and another in the barrel to obtain baseline values. Experiments were performed in an anechoic chamber to minimize extraneous sources of sound. Multiple barrels (not shown) were contained within the same anechoic chamber.

bag) for their sound transmission properties. To reduce external interference from vibration, each fermentation vessel was suspended using rubber bands and the entire experimental system was run in an anechoic chamber (Fig 1). Four different frequency ranges of white noise were tested: 0.2–0.8, 0.8–2, 2–10, and 10–20 kHz, encompassing the entire audible spectrum. Sound intensity was measured directly in the barrel using a hydrophone and compared against sound intensity measured with a hydrophone within each fermentation vessel.

At the lower frequencies (i.e., 0.2–0.8 kHz and 0.8–2 kHz), the soft PVC plastic IV vessel allowed greater sound transmission into the medium than the hard plastic and glass vessels (Table 1). Spectrum plots from sound measurements taken within both the hard plastic and glass vessels exposed to 0.8–2 kHz sound displayed distinguishable peaks under 0.7 kHz, which suggested that those two vessels were more susceptible to interference in sound transmission, or that the vessels were resonating at lower frequencies compared to the soft PVC plastic IV vessel (Fig 2). At the higher frequencies (i.e., 2–10 kHz and 10–20 kHz), the greatest transmission loss was recorded in the soft PVC plastic IV vessel, with overall sound intensity reduced by 7.2–11.1 dBrms re 1 $\mu Pa^2$ @ 1 m (Table 1). In comparison, overall sound intensity was reduced by 0.9–2.9 dBrms re 1 $\mu Pa^2$ @ 1 m for the hard plastic and glass vessels. Across the entire audible spectrum (i.e., 0.2–20 kHz), the average sound intensity loss for the three vessels ranged between 6.6–7.1 dBrms re 1 $\mu Pa^2$ @ 1 m. The soft PVC plastic IV vessels showed the smallest average loss of sound intensity and did not exhibit altered sound profiles at low frequencies (Fig 2 and Table 1), although the hard plastic and glass vessels showed better performance at higher intensities. Furthermore, it is possible to attach leur-lock sampling valves to maintain sterility for the soft PVC plastic bags, reducing the risk of contamination throughout the experiment. Their superior overall sound transmission properties as well as their ability to maintain sterile fermentation conditions during sampling led to us choosing the soft PVC plastic IV bags as the culture vessel for the fermentation experiments.

## Effect of sound frequency on yeast fermentation performance

The underwater sound transmission system, using soft PVC plastic IV bags as fermentation vessels, was used to test the effect of sound frequency and intensity on *S. cerevisiae* fermentation performance. We used the same commercial yeast strain (*Saccharomyces cerevisiae* US05) as our previous underwater sound study [24]. However, we used a defined minimal medium [25], rather than a complex medium, for all sound treatments, although the primary carbon source (maltose at 105 g $L^{-1}$) is consistent with our previous paper [24]. To test the effects of sound frequency, we used the same four frequency bands as used for testing the sound transmission properties of different vessels (i.e., 0.2–0.8, 0.8–2, 2–10, and 10–20 kHz), as well as a no sound or "silence" control. Each band of sound frequency was generated from the underwater speaker at an intensity of 125 dBrms re 1 $\mu Pa^2$ @ 1 m, which is the midrange of speaker output capability. The fermentations, including the silence control, were set up in a large,

**Table 1. Sound transmission loss in three different culture vessels at four different audible spectrum frequency bands.**

| Frequency (kHz) | Baseline (dB$_{rms}$ re 1 $\mu Pa^2$ @ 1 m) | Mean Sound Intensity Difference ± SD (dB$_{rms}$ re 1 $\mu Pa^2$ @ 1 m) | | |
|---|---|---|---|---|
| | | Soft PVC bag | Hard-Plastic Bottle | Glass Flask |
| 0.2–0.8 | 135.23 | - 5.08 ± 2.21 | -17.86 ± 2.54 | -12.92 ± 3.29 |
| 0.8–2.0 | 143.30 | - 2.99 ± 0.06 | - 7.59 ± 0.13 | -10.28 ± 0.05 |
| 2.0–10.0 | 130.55 | - 11.14 ± 1.40 | - 1.82 ± 1.91 | -2.89 ± 0.57 |
| 10.0–20.0 | 130.47 | - 7.18 ± 1.50 | - 0.87 ± 0.45 | - 2.13 ± 0.91 |
| Mean | | - 6.60 | - 7.04 | - 7.04 |

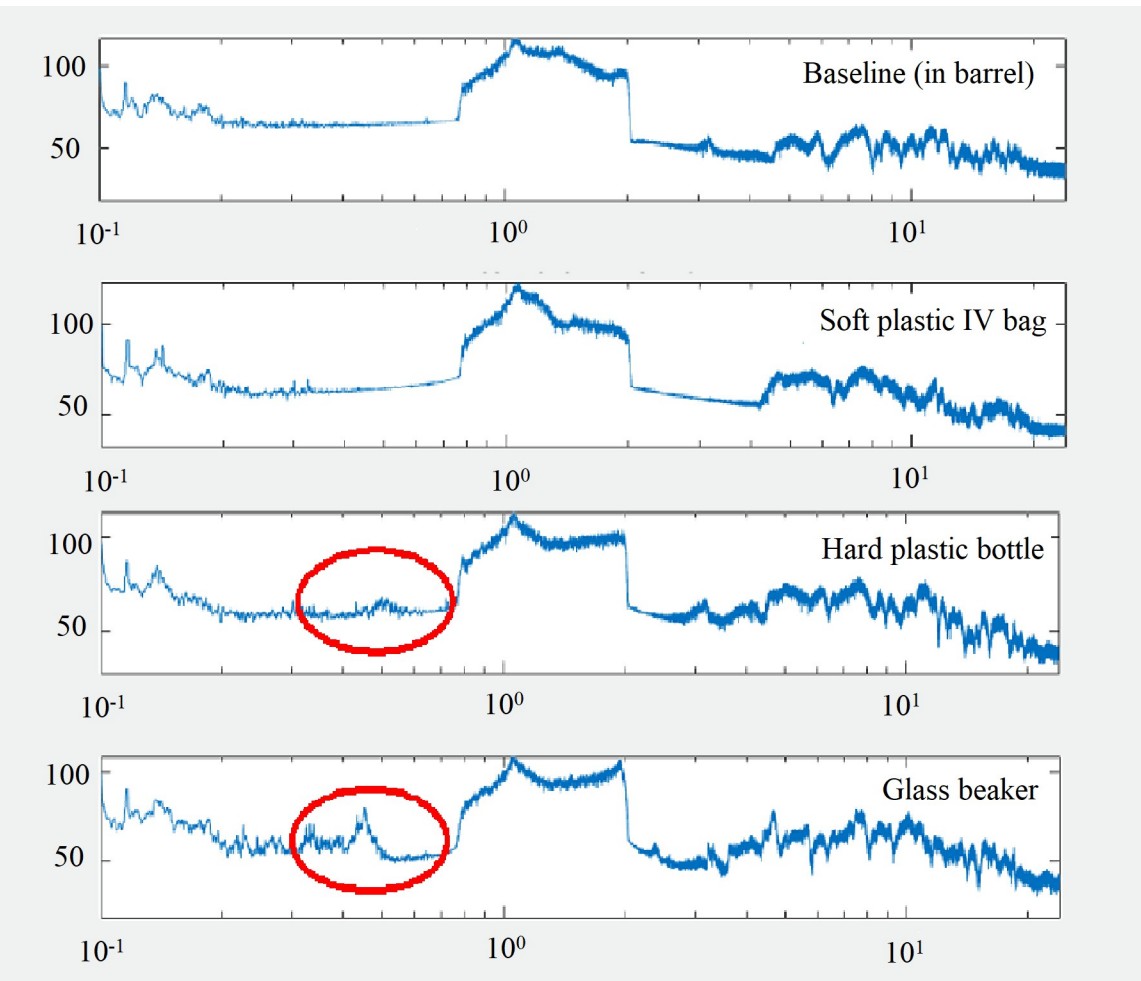

**Fig 2. Comparison of 0.8–2 kHz sound transmission into three different culture vessels.** Sound spectra are plotted as frequency (kHz; x-axis) versus sound intensity (dB re 1 μPa² @ 1 m; y-axis) for baseline in the barrel, and for the soft plastic IV bag, hard plastic bottle, and glass beaker culture vessels following application of 0.8–2.0 kHz white noise. Red circles highlight noticeable sound interference or resonance found for the glass beaker and hard plastic bottle in the 0.2–0.7 kHz range.

acoustically isolated anechoic chamber (University of Auckland) to minimize extraneous noise interference, with a sound baseline of 94.5 dBrms re 1 μPa² @ 1 m. Moreover, the placement of fermentations within the experimental setup was the same for each experimental treatment, with the position of fermentation replicates randomised after each sampling to reduce any biases arising from spatial variation in acoustic intensity [26–28].Fermentation performance was assessed over 60 h at 25 °C in water barrels by measuring specific growth rate, maltose consumption and ethanol production.

The growth in each sound frequency was assessed with an emphasis on the three major growth phases: lag phase, exponential growth phase and stationary phase (Fig 3). Due to contamination, one biological replicate was discarded from the silent treatment. A linear model was developed in R Studio to model the exponential growth phase, with time points excluded from the beginning and end of fermentations considered to be within lag phase or stationary phase respectively. Lag phase length was affected by sound treatment: in silence, very high frequency (10–20 kHz) and high frequency (2–10 kHz) the lag phase lasted 6 hours, whereas in

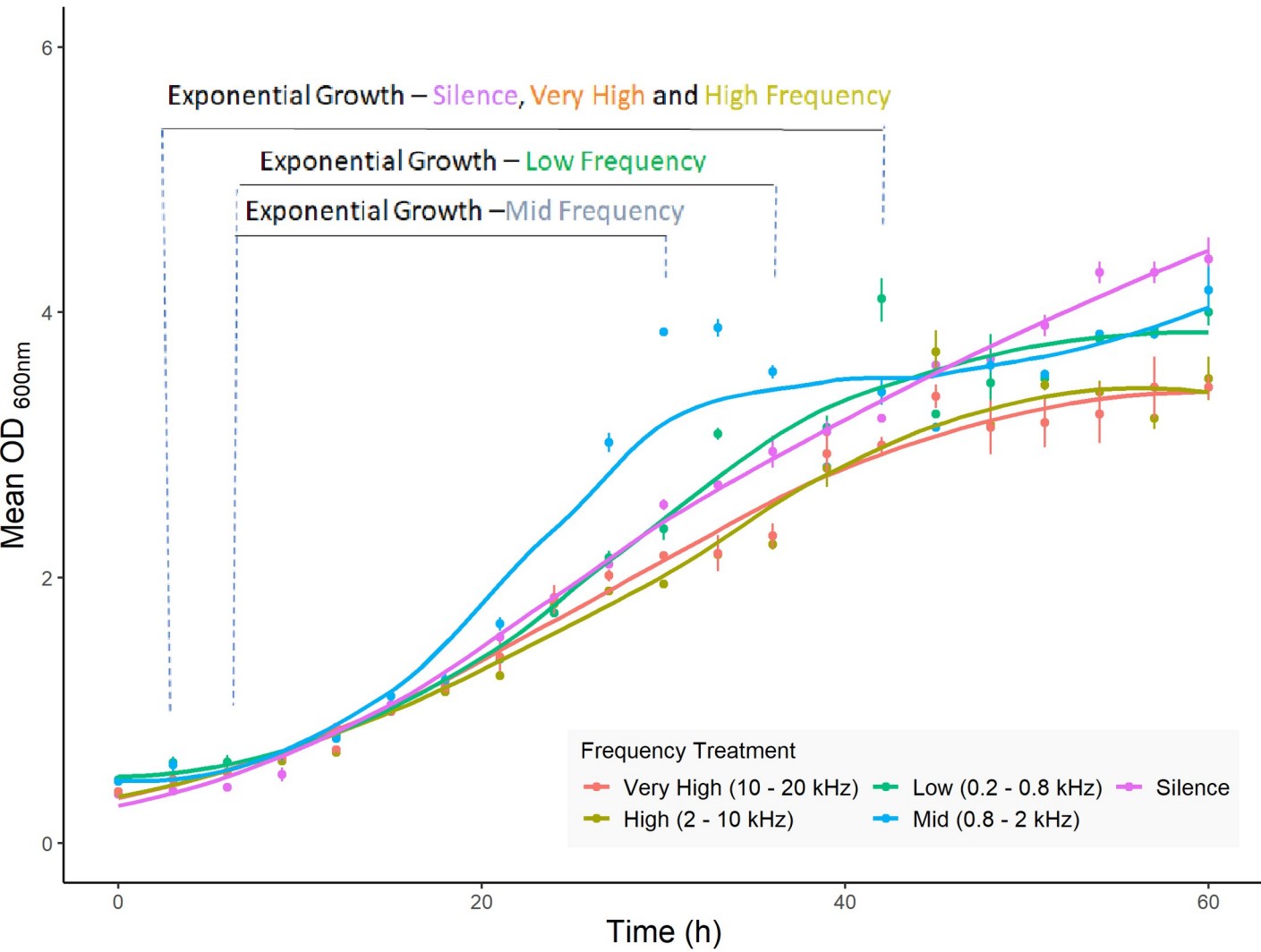

**Fig 3. Growth curves of *S. cerevisiae* US05 exposed to different sound treatments.** Fermentations were performed at 25° C using a minimal medium with 105 g L$^{-1}$ maltose. Biological replicates were subjected to five sound treatments; silence (purple), low frequency (green), mid frequency (blue), high frequency (yellow), and very high frequency (red). Each data point represents the mean of biological replicates (n = 3), with standard deviation represented by error bars. The curves represent the line of best fit for each sound treatment, with the exponential growth phase indicated. Fermentations exposed to low and mid frequency growth had a longer lag phase and shorter exponential growth phase than the other sound treatments and silence. See S1 Table for data used.

mid frequency (0.8–2 kHz) and low frequency (0.2–0.8 kHz) the lag phase was 9 hours (Table 2). Similarly, the duration of the exponential phase was altered depending on the sound treatment: in fermentations exposed to low frequency sound treatment the exponential phase

**Table 2. Growth phases of *S. cerevisiae* US05 in response to sound treatment.**

| Sound Condition | Lag Phase Duration | Exponential Phase Duration |
|---|---|---|
| Silence | 6 hours | 39 hours |
| Low Frequency (0.2–0.8 kHz) | 9 hours | 24 hours |
| Mid Frequency (0.8–2 kHz) | 9 hours | 30 hours |
| High Frequency (2–10 kHz) | 6 hours | 39 hours |
| Very High Frequency (10–20 kHz) | 6 hours | 39 hours |

lasted 24 hours, for mid frequency it was 30 hours, and for silence, high frequency and very high frequency it was 39 hours (Table 2). We also assessed biomass after 60 hours of growth, which was significantly altered among the sound treatments ($p$ value $< 0.05$). The largest biomass was produced in silence whereas the lowest biomass was produced in very high frequency and high frequency sound treatments.

The growth rate ($\mu$), describing growth during the exponential phase alone [29], was calculated for each fermentation replicate using an exponential line of best fit ($R^2 > 0.98$) and subsequently averaged. Low frequency and mid frequency sound treatments exhibited significantly higher growth rates compared to the other treatments ($p$ value $< 0.05$). In particular, the mid frequency treatment had the highest growth rate ($0.079$ hr$^{-1}$), 40.2% higher than that of the silence control (Table 3).

The consumption of maltose was monitored over the course of fermentation. After 60 hours of fermentation, between 52–67% of sugar had been consumed across all five sound treatments (Fig 4A). Overall, fermentations exposed to low frequency and mid frequency sound treatments showed the highest consumption of sugar, consuming significantly more maltose than those in silence, medium or high frequency after 60 hours of growth ($p$ value $< 0.05$). The rate of sugar consumption reflected both the growth rate and the final maltose concentration levels, with the highest consumption ($1.63$ g L$^{-1}$ hr$^{-1}$) occurring in fermentations exposed to the mid frequency sound treatment, 31% higher than the lowest sugar consumption rate ($1.24$ g L$^{-1}$ hr$^{-1}$) observed after exposure to the high frequency sound treatment (Table 2).

The ethanol production throughout fermentation in each sound treatment was investigated every 15 hours. Compared to the growth of *S. cerevisiae*, the production of ethanol took longer to begin, significantly increasing in all sound treatments only after 30 hours. Ultimately the concentration of ethanol reached ~3.2% v/v in all sound treatments, with no significant difference in final ethanol concentration among any of the experimental conditions, or at any time point during fermentation (Fig 4B).

## Effect of sound intensity on yeast fermentation performance

To test the effects of sound intensity on fermentation performance we applied sound at two different intensities: 125 dBrms re 1 $\mu$Pa$^2$ @ 1 m, and 140 dBrms re 1 $\mu$Pa$^2$ @ 1 m, as well as a silence control. We chose to assess the effects of these intensities at a frequency of 0.8–2 kHz, as this was the frequency band with the largest change in growth rate and maltose consumption compared to silence (Table 3). All sound intensity treatments had similar growth curves for the 60 h fermentation period (Fig 5A). The overall the growth rates measured were lower than those measured during the frequency experiment (Table 4). However, the growth curves were all qualitatively similar and there were no significant differences in growth rate between sound intensity treatments or silence. Furthermore, the maltose consumption curves were

**Table 3. Effects of sound frequency on specific growth rates and maximum maltose consumption rates of *S. cerevisiae* fermentations.**

| Frequency | Mean growth rate ($\mu$) ± SD | Maximum maltose consumption rate (g L$^{-1}$ h$^{-1}$) ± SD |
|---|---|---|
| Silence | 0.056 ± 0.002 | 1.25 ± 0.02 |
| Low Frequency (0.2–0.8 kHz) | 0.064 ± 0.003 | 1.63 ± 0.08 |
| Mid Frequency (0.8–2 kHz) | 0.079 ± 0.003 | 1.59 ± 0.05 |
| High Frequency (2–10 kHz) | 0.051 ± 0.002 | 1.24 ± 0.12 |
| Very High Frequency (10–20 kHz) | 0.052 ± 0.002 | 1.36 ± 0.07 |

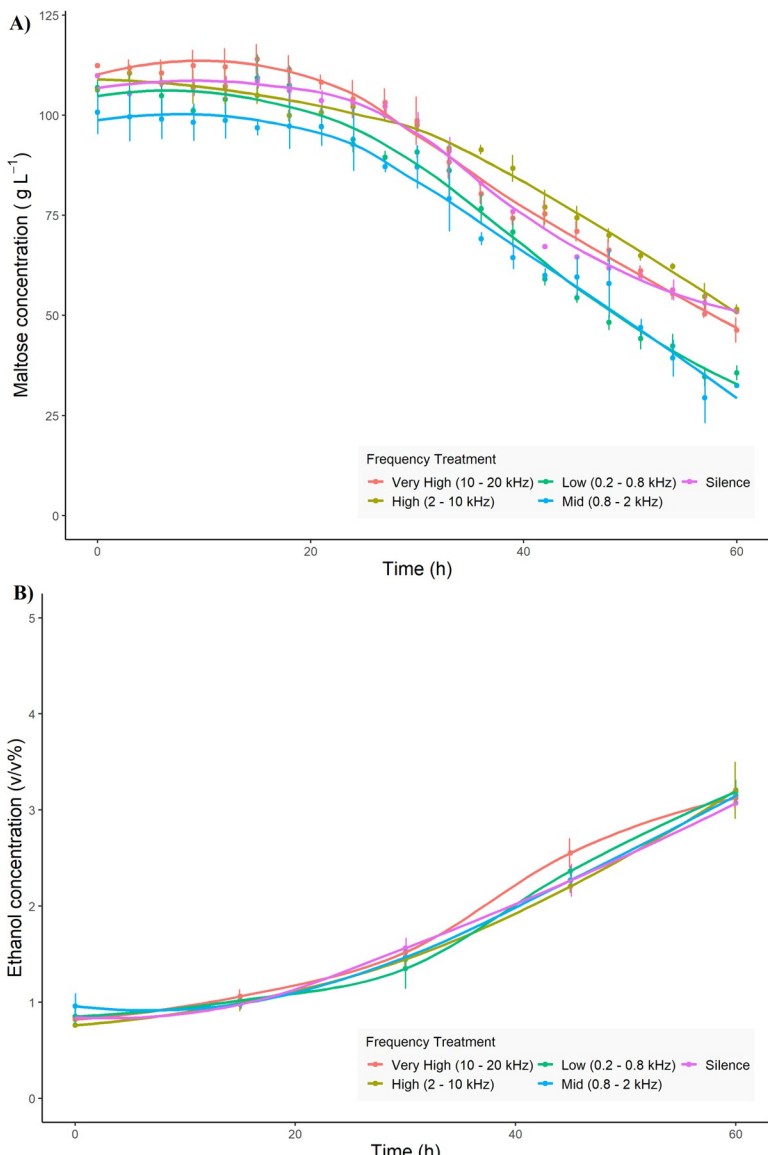

**Fig 4. Characterization of *S. cerevisiae* US-05 fermentation parameters following exposure to different frequency sounds.** A) Maltose concentration (g L$^{-1}$) every 3 h, and B) ethanol concentration (v/v%) every 15 h, were quantified throughout the 60 h fermentation. The line of best fit is plotted for each sound treatment. Data points represent the mean of biological replicates (n = 3), with error bars showing standard deviation. See S1 Table for data used.

very similar among all intensity treatments (Fig 5B), with no significant differences in maximum maltose consumption rate identified. Similarly, ethanol production was similar throughout the 60 h fermentation period for all sound treatments (Fig 5C). Together, these results suggest there is little effect of sound intensity treatment on fermentation performance at the selected frequency band under the experimental conditions employed.

The 125 dBrms re 1 µPa$^2$ @ 1 m and silence treatments from the sound intensity experiment are effectively repeats of the 0.8–2 kHz and silence treatments from the prior frequency experiment. However, while we observed significantly higher growth and maltose consumption rates compared to the silence for the 0.8–2 kHz at 125 dBrms re 1 µPa$^2$ @ 1 m treatment in the sound intensity experiment, these differences were not apparent in the intensity experiment. Indeed, maltose

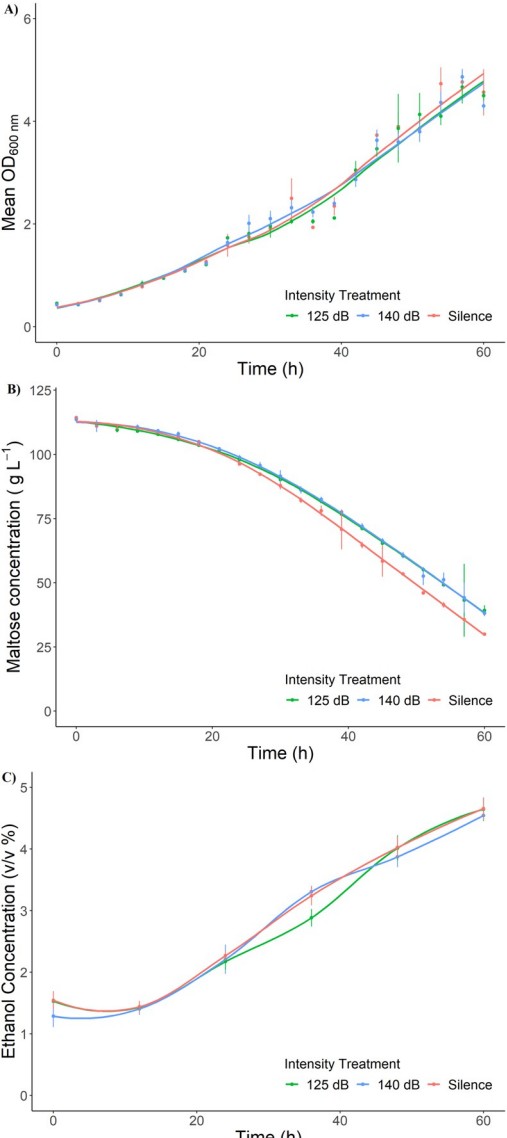

**Fig 5. Effect of sound intensity on fermentation performance of *S. cerevisiae* in minimal medium.** Plots show yeast growth (A), maltose consumption (B), and ethanol production (C) over 60 h for two sound intensities at 0.8–2 kHz, as well as silence. Mean values (dots) and standard deviation (error bars) for three biological replicates are shown. See S1 Table for data used.

consumption was actually higher (although not significantly so) in silence compared to sound treatment in the intensity experiment. As a consequence, while the initial frequency experiment

**Table 4. Effects of sound intensity on specific growth rates of *S. cerevisiae* fermentations.**

| Intensity (dBrms re 1 μPa$^2$ @ 1 m) | Mean specific growth rate (μ) ± SD | Maximum maltose consumption rate (g L$^{-1}$ h$^{-1}$) ± SD |
|---|---|---|
| 125 | 0.0446 ± 0.0006 | 1.612 ± 0.011 |
| 140 | 0.0434 ± 0.0005 | 1.549 ± 0.013 |
| Silence | 0.0446 ± 0.0006 | 1.721 ± 0.040 |

suggested that sound produces a significant biological effect on fermentation, the combination of both experiments fails to find any compelling evidence for a consistent effect of sound application on yeast fermentation performance under the standardized experimental conditions.

## Discussion

In this study we used an experimental platform that was designed to control delivery of sound to determine the effect of audible white noise sound on *S. cerevisiae* fermentations. To do this, we used speakers submerged in a water barrel in which the culture vessel was also submerged, making the densities of the media through which sound passed similar. By measuring sound directly within the culture vessel, we were able to show that while some change in sound still occurred between the speaker and fermentation vessel, our setup was effective in minimizing these changes, particularly for the soft-bag culture vessel we selected. In addition, rather than reducing background interference through the use of soundproofing material(s) attached to the walls of a testing chamber, as is typically employed [7, 22], we performed our fermentations in a large anechoic chamber to maximize acoustic isolation of the experiment [18].

Our results showed some impact of sound on yeast growth rate and maltose consumption, but not ethanol production, when different sound frequencies were compared. However, we did not see any significant effect of sound treatment when different sound intensities were compared. Thus, the significant effects observed under the frequency experiment were not replicated with the intensity experiment leading us to conclude that audible sound does not have a clear biological effect on *S. cerevisiae* minimal medium fermentations under the experimental conditions of this study.

Our results contrast most published microbial studies, where audible sound is commonly reported to increase growth and/or biomass production compared to a silent control [5–12, 22, 23], although some studies have found a negative impact on growth [30, 31]. The discrepancy with our study could be explained by differences in the species/strain used and/or the culturing system employed. However, another potential source of difference is the sound parameters. We used white noise sound bands covering the entire audible spectrum, where the power spectrum of each band was adjusted such that the average sound intensity within the barrel was equally distributed across frequencies within each band. This was to try and capture responses that might occur with specific frequencies, of which there are infinitely many that could be individually tested. However, it is possible that these broad frequency bands actually masked effects seen in experiments using pure tone sound stimuli. (20 Hz to 20 kHz). Moreover, some studies have found responses to music [11, 32–34]. Music, while covering multiple frequencies, involves periodic sound stimuli that are more similar to the periodic sounds microorganisms are likely to encounter in nature than the continuous sounds we used. If microorganisms have evolved responses to environmental sounds, the periodic nature of these sounds may be an important factor in producing responses. Our results also contrast studies that have reported effects from the application of ultrasound sonication, including effects on *S. cerevisiae* during fermentation [35, 36]. It will be interesting to determine whether this is because of differences between audible sound and ultrasound, or because of other causes.

[35, 36] An alternative explanation for the differences in results is that by removing the air-to-liquid route of sound transmission that other studies have employed, we may have removed the stimulus that is producing the previously reported sound effects. For example, external application of sound through air may cause fermentation vessels to vibrate rather than the sound pressure being transferred directly into the fermentation vessel as occurred in this study. This explanation is consistent with our initial study using our underwater audible sound transmission system, which found little effect on sugar utilization or volatile organic

compound production during *S. cerevisiae* beer wort fermentation [24]. If this explanation is correct, our results could be used to help identify the critical aspects of audible sound that are responsible for influencing microbial growth. Therefore, we suggest that testing of the difference between liquid and air sound transmission is an important next step in trying to better understand the impact of sound on microbial growth.

Our results also show the importance of full experimental replication of results, as this showed that what appeared to be significant differences in fermentation parameters as a consequence of sound treatment from our first set of experiments are likely spurious results. Given that previous studies typically have not replicated their experiments in the way we have here, it is possible that some of the effects of sound they report may also be spurious. Essentially what we are reporting here, then, is a batch effect, which account for most of the variation in industrial scale fermentations [37]. Critically, the magnitude of this batch effect variation must be sufficient to mask the true impact of sound (if any exists) as we did not see consistent results between replicates. The minimal medium used for all the experiments here was prepared from identical stock solutions and the experiments were performed in an anechoic chamber, therefore we think batch-to-batch variation is likely to be minimal, suggesting that any impact on sound on *S. cerevisiae* fermentation must also be minimal. This is particularly relevant for the use of sound in industrial applications, as any differences caused by sound would need to be large enough to overcome batch effects.

In summary, using a carefully controlled experimental system for delivering audible sound to microbial liquid cultures, we find little difference in the key measured parameters for *S. cerevisiae* minimal medium fermentations performed in an anechoic chamber as a consequence of audible sound application. We suggest that standardized sound application systems that emphasize the sound received by the organisms, such as we have established, are essential for determining what impacts audible sound has on microbial cultures, and for enabling comparisons between different studies. These measures should then enable us to gain a comprehensive understanding of the nature of the sound stimulus that produces effects on microbial growth and metabolism, as well as ultimately providing insight into any mechanism that mediates these effects.

## Materials and methods

### Sound composition and transmission

Four sound source files were generated and stored as .wav files in MATLAB (v. R2018b; Math Works, USA) for this study. The four sound source files consisted of white noise of equal sound power across a defined frequency band i.e., 0.2–0.8 kHz, 0.8–2 kHz, 2–10 kHz, and 10–20 kHz (Fig 6). Sound source files of 3 min duration were played on a continuous loop using VLC player (Videolan). The input signal was amplified by a 1000 W power amplifier (Pioneer Gm-A6704 A Series) to drive each underwater speaker, with mean sound output being adjusted using the volume function.

Sound frequency and intensity underwater was measured using a calibrated HTI-96-Min broadband hydrophone (High Tech, USA) with a flat frequency response over the audible frequency range. Outputs were recorded for 10 sec onto a digital recorder (R-05 Recorder, Roland, Japan). A subset of this recording (4–8 sec) was analysed in MATLAB using bespoke code to calculate mean sound intensity and sound frequency composition (S1 Appendix).

### Sound transmission testing

Three vessel types: Nalgene 1000 mL plastic bottle, Glassco 500 mL round bottom flask, and 600 mL polyvinylchloride (PVC) IV feeding bag (Henso Medical), were filled with fresh water and suspended by acoustic isolating rubber bands in a 115 L plastic barrel filled with fresh

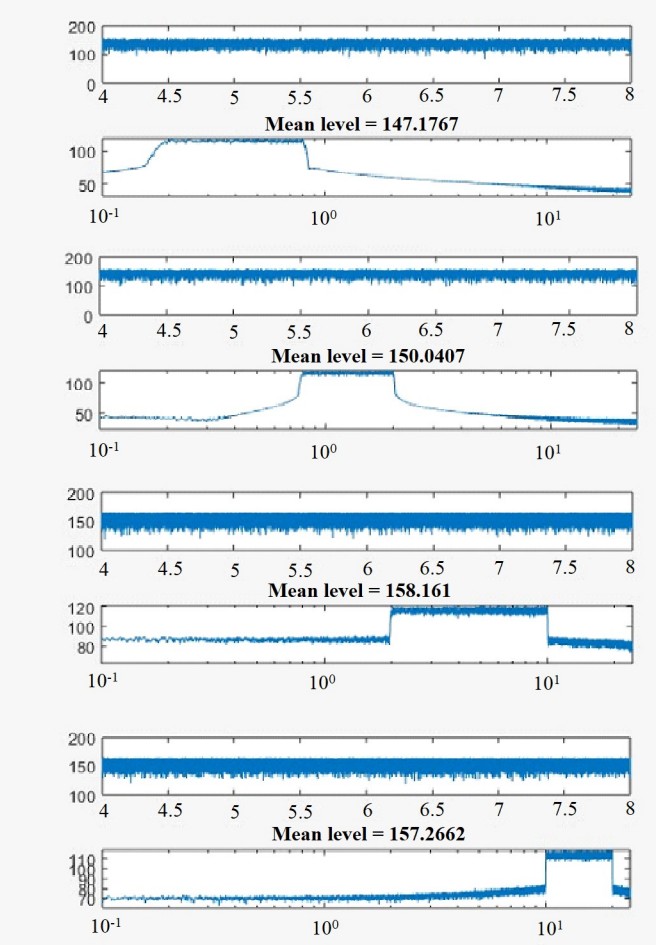

**Fig 6. Spectrograms for white noise in four frequency bands.** The pairs of graphs are: at top, time (sec; x-axis) versus sound intensity (dB re 1 $\mu Pa^2$ @ 1 m; y-axis) used to calculate mean sound intensity (dB$_{rms}$ re 1 $\mu Pa^2$ @ 1 m); and at bottom, sound frequency in Hz (x-axis) versus sound intensity (dB re 1 $\mu Pa^2$ @ 1 m). Results are shown for the four frequency ranges used in this study: 0.2–0.8 kHz (A); 0.8–2 kHz (B); 2–10 kHz (C); and 10–20 kHz (D). Mean sound intensity level across the entire frequency bandwidth is indicated for each frequency band.

water. Mean sound intensity inside each suspended vessel was recorded and compared against the mean sound intensity measured in the barrel without any vessels present.

## Strains and culturing methods

*Saccharomyces cerevisiae* US-05 ale strain (Fermentis, USA) isolated from a commercial package was maintained as a stock at -80 ˚C and revived at 25˚C on PDA plates. Minimal medium contained 105 g L$^{-1}$ maltose, 10 g L$^{-1}$ $(NH_4)SO_4$, 3 g L$^{-1}$ $KH_2PO_4$, 1.5 g L$^{-1}$ $MgSO_4$, vitamins, and trace metals [25]. For ferments, 12 mL of 500 mg L$^{-1}$ ergosterol (dissolved in EtOH) was added to 500 mL of medium after autoclaving to obtain a final concentration of 6 mg L$^{-1}$. Liquid medium was prepared in bulk prior to each fermentation and distributed evenly between replicates.

## Sound treatment and sampling

The experimental setup (Fig 7) consisted of; a submerged speaker, 115 L plastic water barrel and 1 L light plastic IV bags in which the fermentations were conducted. For each sound

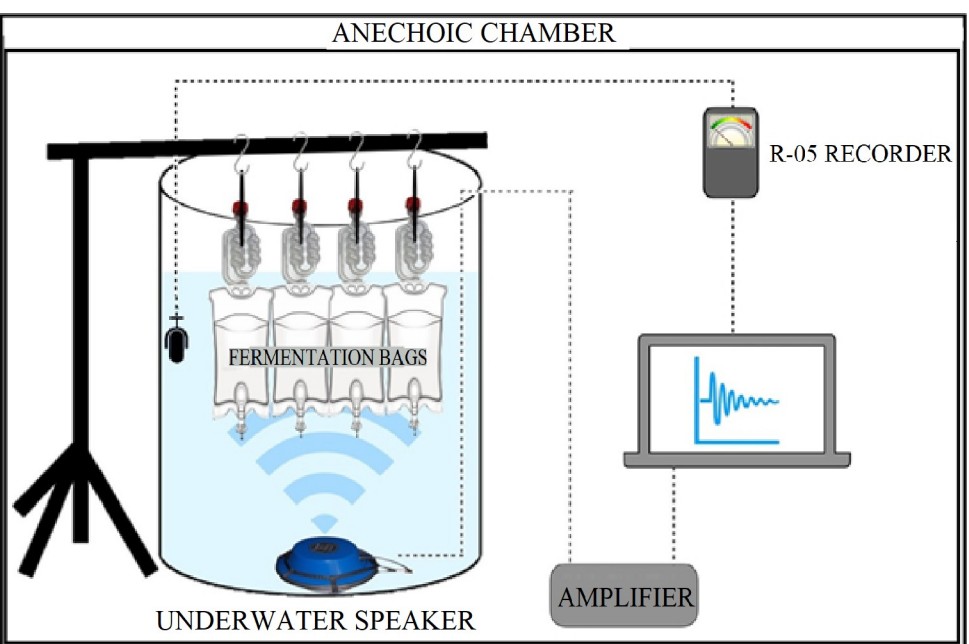

**Fig 7. Sound experimental setup.** Fermentation vessels were suspended, and acoustically isolated, from a microphone stand by rubber bands. These vessels were submerged in a 115 L plastic barrel. For each sound treatment, three biological replicates and one uninoculated control were suspended within the barrel. Sound treatments were applied to fermentations using an underwater speaker, connected to an amplifier and computer. The sound volume was recorded using a calibrated hydrophone and adjusted using the computer volume control to the appropriate intensity. The entire experimental setup was placed within the University of Auckland's anechoic chamber. Multiple barrels (not shown) were able to be placed within the anechoic chamber simultaneously. Diagram is not to scale.

treatment three fermentations (n = 3) were conducted, with a further uninoculated IV bag included to check for contamination. The IV bags had sampling ports (One-Way Stopcock Valve Female Luer to Male Luer) attached to the bottom, and an airlock attached at the top. Sampling ports and airlocks were sterilized by submerging in 70% ethanol for 24 hours. Each bag was suspended by rubber bands from an acoustically isolated stand. The speaker used depended on the specific sound being played; frequencies under 800 Hz used a J9 underwater speaker (Underwater Sound Reference Detachment, National Research Laboratories USA); frequencies between 800–2000 Hz used an underwater speaker (LL916C-050, Lubell Labs); and frequencies between 2–20 kHz used a 4-inch midrange speaker (CW2190, Jaycar NZ) inside two light plastic bags. The speaker was connected to a computer via a 1000 W power amplifier (Pioneer Gm-A6704 A Series) with the sound power output adjusted using the computer and VLC volume functions. Each sound file (.wav) was played back using VLC media player.

Prior to sound treatment viable cells were produced in a pre-inoculum. The pre-inoculum for each experiment was started using four-day old colonies grown on PDA. Three loops of yeast were transferred to 1 mL of sterile saline solution (0.9% w/v) and vortexed. Subsequently the saline/yeast suspension was added to 250 mL of minimal medium and incubated aerobically at 25°C and 160 RPM for ~43 hours. Depending on the number of cells required, multiple pre-inoculum cultures were grown simultaneously. Once the pre-inoculum reached late exponential-phase (an $OD_{600}$ of 2–3.5), the pre-inoculum(s) were collected and mixed. This combined pre-inoculum was then divided into aliquots containing $4 \times 10^{11}$ cells, which was calculated to give a final pitching rate of $8 \times 10^8$ cells/mL. Cells

were pelleted at $14,000 \times g$ for 5 min and resuspended in 50 mL minimal medium. Sterile 600 mL IV bags with 450 mL of minimal medium were inoculated with this cell suspension (giving a final fermentation volume of 500 mL), submerged in the water barrel, and maintained at 25˚C in an anechoic chamber for 60 h.

Fermentations were exposed to five different white noise sound treatments; low frequencies (0.2–0.8 kHz), mid frequencies (0.8–2 kHz), high frequencies (2–10 kHz), very high frequencies (10–20 kHz) and silence (no sound input). The power spectrum of each band was adjusted such that the average sound intensity within the barrel was 125 $dB_{RMS}$ re 1 $\mu Pa^2$ @ 1m, with power equally distributed across frequencies within each band, to the limit of the speaker. The sound treatments were played continuously for the duration of the experiment, with sound files looped in VLC media player every 3 min. Subsequently, different fermentations were exposed to three sound intensity treatments, 125 dBrms re 1 $\mu Pa^2$ @ 1 m, and 140 dBrms re 1 $\mu Pa^2$ @ 1 m, and silence, with a frequency of 0.8–2 kHz.

Every 3 h following inoculation the fermentation bags were shaken to resuspend yeast, ~5 mL of ferment was drawn through a sterile port, and the optical density (600 nm) measured using a spectrophotometer (1 cm path cuvette; Biochrom, Cambridge, England). Where $OD_{600} > 0.8$, samples were diluted with fresh medium to a suitable level before reading. Maltose quantification was performed using a dinitrosalicylic acid (DNS) colorimetric assay [38] or a DSR-Λ refractometer (SCHMIDT + HAENSCH, Germany). Refractive index outputs were converted into g $L^{-1}$ following a standard curve conversion (Mettler Toledo). Ethanol was measured from 1 mL samples of ferments taken every 12 h using gas chromatography-mass spectrometry, as previously described [39]. Absolute ethanol concentrations were calculated by normalizing against D4-methanol present in all samples and comparing the relative abundance of ethanol in each sample to known ethanol v/v% standards via a linear standard curve (also normalised to D4-methanol) covering v/v%.ethanol concentrations of 0.4, 1, 2, 3, 4, and 8.

### Statistical analysis

All data processing was carried out in R Studio Version 1.1.414. Growth rates, maltose consumption rates and ethanol production rates were compared with a standard univariate analysis of variance (ANOVA) test with a post-hoc Tukey HSD correction employed to correct for multiple comparisons.

### Growth models

To model the growth of *S. cerevisiae* throughout fermentation, a lag phase, an exponential growth phase, and a final stationary phase were assumed. The $OD_{600}$ values for each biological replicate, in each sound treatment, were log transformed and plotted against time. The exponential growth phase duration was then determined through a trial-and-error method in which the data were fitted to a linear model. Time points included in the linear model were trimmed from the beginning and end of fermentation until a maximal adjusted R squared value ($>0.91$) was obtained for each fermentation. Time points excluded were considered the lag and stationary phases. To determine the growth rate ($\mu$) for each sound treatment the coefficient of time ($hr^{-1}$) was extracted from each linear model and the mean taken between biological replicates.

### Maltose consumption

To investigate maltose consumption the concentration of maltose (g $L^{-1}$) was plotted against time for each fermentation in each sound treatment, and a linear model was built to identify

the maximal rate of consumption for each fermentation. Time points at the beginning and end of fermentations were trimmed to optimise the adjusted R squared values ($> 0.92$). From each linear model the coefficient for time ($g\,L^{-1}\,hr^{-1}$) was extracted and the mean taken between biological replicates of each sound treatment.

### Ethanol production

To compare the rate of ethanol production between sound treatments ethanol concentration (v/v %) was plotted against time for each fermentation in each sound treatment, the initial time point of each fermentation was trimmed from the dataset, and a linear model was constructed. The coefficient of time ($v/v.hr^{-1}$) was extracted, and the mean taken between biological replicates for each sound treatment.

## Supporting information

**S1 Table. Tables providing the underlying raw yeast growth, maltose concentration and ethanol concentration data used to produce Figs 3–5.** Yeast data are provided as $OD_{600nm}$. Maltose concentration is $g.L^{-1}$. Ethanol concentration is % v/v, with the concentration being determined the same way in both cases.
(XLSX)

**S1 Appendix. Provides the MATLAB code used to calculate mean sound intensity and sound frequency from the sound recordings.**
(DOCX)

## Acknowledgments

We thank Sylvie Hermann-Le Denmat (University of Auckland) for technical assistance, and Michael Kingan, Gian Schmid and Andrew Hall (University of Auckland) for kindly providing assistance with the anechoic chamber.

## Author Contributions

**Conceptualization:** Alastair Harris, Pat Silcock, Graham Eyres, Silas G. Villas-Bôas, Andrew Jeffs, Austen R. D. Ganley.

**Data curation:** Rachel Benitez, Alastair Harris.

**Investigation:** Rachel Benitez, Alastair Harris, Evie Mansfield.

**Methodology:** Rachel Benitez, Alastair Harris, Silas G. Villas-Bôas.

**Resources:** Andrew Jeffs, Austen R. D. Ganley.

**Supervision:** Andrew Jeffs, Austen R. D. Ganley.

**Validation:** Rachel Benitez, Alastair Harris, Evie Mansfield, Silas G. Villas-Bôas.

**Writing – original draft:** Rachel Benitez, Alastair Harris.

**Writing – review & editing:** Rachel Benitez, Alastair Harris, Evie Mansfield, Pat Silcock, Graham Eyres, Silas G. Villas-Bôas, Andrew Jeffs, Austen R. D. Ganley.

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
