## [Decision Letter · Decision Letter 0]

26 Oct 2022

PONE-D-22-27564Direct liquid transmission of sound has little impact on fermentation performance in Saccharomyces cerevisiaePLOS ONE

Dear Dr. Ganley,

Thank you for submitting your manuscript to PLOS ONE. After careful consideration, we feel that it has merit but does not fully meet PLOS ONE’s publication criteria as it currently stands. Therefore, we invite you to submit a revised version of the manuscript that addresses the points raised during the review process.

We look forward to receiving your revised manuscript.

Kind regards,

Shashi Kant Bhatia

Academic Editor

PLOS ONE

Journal Requirements:

"We thank Sylvie Hermann-Le Denmat (University of Auckland) for technical assistance, and Michael Kingan, Gian Schmid and Andrew Hall (University of Auckland) for kindly providing assistance with the anechoic chamber. This work was supported by a Smart Ideas grant from the New Zealand Ministry for Business, Innovation, and Employment [UOAX1713]."

"This work was supported by a Smart Ideas grant from the New Zealand Ministry for Business, Innovation, and Employment [UOAX1713; https://www.mbie.govt.nz/science-and-technology/science-and-innovation/funding-information-and-opportunities/investment-funds/endeavour-fund/] to ARDG, AJ, SV-B, PS and GE. he funders had no role in study design, data collection and analysis, decision to publish, or preparation of the manuscript."

Reviewers' comments:

Reviewer's Responses to Questions

**Comments to the Author**

1. Is the manuscript technically sound, and do the data support the conclusions?

Reviewer #1: Partly

Reviewer #2: Partly

2. Has the statistical analysis been performed appropriately and rigorously? 

Reviewer #1: Yes

Reviewer #2: Yes

3. Have the authors made all data underlying the findings in their manuscript fully available?

Reviewer #1: Yes

Reviewer #2: Yes

4. Is the manuscript presented in an intelligible fashion and written in standard English?

Reviewer #1: Yes

Reviewer #2: Yes

5. Review Comments to the Author

Reviewer #1: Overall, I find the manuscript technically sound with data supporting the conclusions. It is however important to point out that the study is performed with white noise signals and thus the conclusions drawn are valid for this class of signals. Please clarify this in abstract, discussion, conclusions etc.

I would also appreciate a rationale and/or discussion for the choice of white noise signals as stimuli. Many if not most mechanical (including acoustic) stimuli in nature are of periodic nature. For instance, is the studied organism S. cerevisiae reported to oscillate at a frequency around 1 kHz (see e.g. Pelling et al., Science. 2004 Aug 20;305(5687):1147-50). Also, some of the provided references with contradicting results used music signals as stimulary sources.

Minor comment: Ergosterol stock solution to growth medium. Please clarify if the ergosterol addition was 12 mL TOTAL or 12 mL per L medium.

Reviewer #2: Were the anechoic chamber dimensions similar in control as it was in the test experiments? Moreover, the material and placement of the fermentation bags/vessels should be same for all control and test experiments. Spatial variations in case of sound waves are known to alter the acoustic intensity and may result to artifacts due to fluctuating intensity (Sutkar and Gogate, 2009, Moholkar et al., 2000; Gogate et al., 2002).

Sutkar, V.S., Gogate, P.R., 2009. Design aspects of sonochemical reactors: techniques for understanding cavitational activity distribution and effect of operating parameters. Chem. Eng. J. 155, 26–36.

Moholkar, V.S., Sable, S.P., Pandit, A.B., 2000. Mapping the cavitation intensity in an ultrasonic bath using the acoustic emission. AIChE J. 46, 684–694.

Gogate, P.R., Tatake, P.A., Kanthale, P.M., Pandit, A.B., 2002. Mapping of sonochemical reactors: review, analysis, and experimental verification. AIChE J. 48, 1542–1560.

-Why did author not consider testing the sound beyond the range they have included as the enhancement in ethanol titer has not been found here. There are some interesting observations after mechanistic investigations and more enhancement in the studies where fermentation was performed under intermittent ultrasonic irradiation. Along with their results, authors can discuss the findings in other sound-intensified ethanol fermentation [Singh et al. 2015, Ultrasonics sonochemistry 21 (1), 200-207; Singh et al. 2015, Bioresource Technology 188, 287-294; Sulaiman et al. 2021, Biochemical Engineering Journal 54 (3), 141-150)]

6. PLOS authors have the option to publish the peer review history of their article (what does this mean?). If published, this will include your full peer review and any attached files.

Reviewer #1: No

Reviewer #2: No

---

## [Author Response · Author response to Decision Letter 0]

9 Jan 2023

Please see the responses to reviewers document that was uploaded as part of this submission.

---

## [Decision Letter · Decision Letter 1]

1 Feb 2023

Direct liquid transmission of sound has little impact on fermentation performance in Saccharomyces cerevisiae

PONE-D-22-27564R1

Dear Dr. Ganley,

We’re pleased to inform you that your manuscript has been judged scientifically suitable for publication and will be formally accepted for publication once it meets all outstanding technical requirements.

Kind regards,

Shashi Kant Bhatia

Academic Editor

PLOS ONE

---

## [Editor Report · Acceptance letter]

7 Feb 2023

PONE-D-22-27564R1 

Direct liquid transmission of sound has little impact on fermentation performance in *Saccharomyces cerevisiae*

Dear Dr. Ganley:

I'm pleased to inform you that your manuscript has been deemed suitable for publication in PLOS ONE. Congratulations! Your manuscript is now with our production department. 

Kind regards, 

on behalf of

Dr. Shashi Kant Bhatia 

Academic Editor

PLOS ONE